# Bio-inspired creation of heterogeneous reaction vessels via polymerization of supramolecular ion pair

Ke Dong[1], Qi Sun[2], Yongquan Tang[1], Chuan Shan [3], Briana Aguila [3], Sai Wang[1], Xiangju Meng[1], Shengqian Ma [3] & Feng-Shou Xiao[1]

Precise control of the outer-sphere environment around the active sites of heterogeneous catalysts to modulate the catalytic outcomes has long been a challenge. Here, we demonstrate how this can be fulfilled by encapsulating catalytic components into supramolecular capsules, used as building blocks for materials synthesis, whereby the microenvironment of each active site is tuned by the assembled wall. Specifically, using a cationic template equipped with a polymerizable functionality, anionic ligands can be encapsulated by ion pair-directed supramolecular assembly, followed by construction into porous frameworks. The hydrophilic ionic wall enables reactions to be achieved in water that usually requires organic solvents and also facilitates the enrichment of the substrate into the hydrophobic pocket, leading to superior catalytic performances as demonstrated by the industrially relevant hydroformylation. Remarkably, the formation of the supramolecular assembly and catalyst encapsulation further engenders reaction selectivity, which reaches an even greater extent after construction of the porous framework.

[1] Key Lab of Applied Chemistry of Zhejiang Province, Zhejiang University, 310028 Hangzhou, China. [2] College of Chemical and Biological Engineering, Zhejiang University, 310027 Hangzhou, China. [3] Department of Chemistry, University of South Florida, Tampa, FL 33620, USA. Correspondence and requests for materials should be addressed to Q.S. (email: sunqichs@zju.edu.cn) or to S.M. (email: sqma@usf.edu) or to F.-S.X. (email: fsxiao@zju.edu.cn)

uter-sphere design is customary for biomolecules to control chemical reactivity but is a rarity for traditional synthetic systems[1–3]. With the aim of developing synthetically useful catalysts, structures that mimic enzyme reactivity, in which a cavitand is connected to an active site, have been a pioneering research for decades[4–9]. The development of supramolecular assemblies is one of the most promising strategies for encapsulating catalytically relevant elements in a confined space. This allows for their performance to be regulated by rational design of the assembly host rather than merely relying on inner-sphere ligands, reminiscent of enzymes with regard to catalytic acceleration by proximity effects in pocket[10–20]. While success has been achieved using these systems in the context of homogeneous catalysis, to meet the challenges associated with product purification and catalyst reusability, tethering these supramolecular catalysts on insoluble carriers would allow for an increase in operational ability[21]. Following these, we envisioned that if an assembly can be constructed into a highly porous framework in a self-replication manner, this may capture the advantages of both worlds, as they comprise a solid catalyst that matches the high selectivity and activity of state-of-the-art homogeneous catalysts, and also demonstrate robust stability in processes. The performance of the individual catalytic units, in turn, may be further promoted after the construction of framework materials by virtue of their merits, such as the high surface area, and thus increased interfacial area between the multiple phases involved in the reaction[22–25].

To validate of this hypothesis, we were motivated by the exploration of templated assembly supramolecular encapsulation, given the proven efficiency[26,27]. Among the developed template assembly strategies, ion pair-directed assembling has received considerable interest[27–31], due to the following attributes: (i) the ubiquity of ionic compounds and thereby the potential large catalyst libraries; (ii) their small-molecule nature enabling ready modification to install functionalities for the potential construction of porous frameworks, as well as providing an amenable route for rigorous control of the reactivity; (iii) the hydrophilic ionic outer-sphere and hydrophobic ligand pocket of the resulting assembly allowing for catalysis in the aqueous phase or at the organic/aqueous phase interface; (iv) the created hydrophobic pocket serving as a trap for enriching reagents in the course of a reaction in the aqueous phase and thus the overall efficiency.

With these considerations in mind, we describe here the utility of ion pair-directed supramolecular assembly as a building unit in the creation of related catalytically active porous frameworks. To demonstrate the proof-of-concept, we choose quaternary phosphonium as the cationic template, given its chemical stability, amenable synthesis, together with easy-to-get properties of anionic ligands[32–34]. To expand the applicability, we further construct the resulting supramolecular assemblies into porous frameworks (Fig. 1). The high accessibility of the open coordination site in the frameworks could be harnessed for catalytic reactivity with metal species. Such supramolecular assembly-derived materials combine the advantages of a porous material and the reactivity of embedded organometallic species while being imparted with additional interesting properties stemmed from the unique assembly structure, thereby showing exceptional performances in the industrially relevant hydroformylation reaction. Due to the imparted hydrophilic outer-sphere environment, the reactions can proceed in aqueous solutions, allowing for ready product separation. Given the versatility, we envisage the strategy presented herein as a cornerstone to realize a new paradigm in the design of catalytic materials.

## Results

**Material synthesis**. To implement this strategy, we sought to employ a vinyl functionalized quaternary phosphonium salt (V-QP), (4-vinyl-benzyl)-tris-(4-vinyl-phenyl)-phosphonium chloride, as a cationic template with the following considerations: in view of various advanced materials developed, porous organic polymers are at the frontier due to their modularity and high stability[35–42], whereby permanently charged microporous networks have been employed for numerous applications[22–25]. Moreover, with regard to their synthesis, a free radical polymerization strategy was adopted given its lack of reactants or side products in conjugation with its excellent monomer tolerance[35]. The equipped vinyl groups in the template provide the needed polymerization sites for the construction of porous frameworks, while the phosphonium site is essential to the supramolecular assembly process. We introduced the ionic template strategy to modify the out-sphere environment around the ligand-binding sites. Upon mixing the above cationic compound and a certain equivalent of sodium salt of sulfonated ligands in dimethylformamide (DMF), with equal moles of the chloride anions and sodium cations in the final composition, the assemblies were formed by electrostatic interactions, where various ligands with open coordination sites bind to the phosphonium walls (Table 1). The structures of the resulting assemblies were well characterized by means of multinuclear NMR spectroscopy and Electrospray ionization mass spectrometry (ESI-TOF-MS, Supplementary Figs. 1–6). The resulting supramolecular assemblies are glass-like amorphous solids (Supplementary Fig. 7). These assemblies were quantitatively transformed into the corresponding porous framework materials by polymerization of the vinyl groups on the assemblies under solvothermal conditions in DMF at 100 °C for 24 h with the assistance of a free radical initiator azobisisobutyronitrile (AIBN), resulting in powder-like materials that are insoluble in conventional solvents. Here, three sodium salts of the sulfonated ligands, triphenylphosphine (PPh₃), 4,5-bis(diphenylphosphino)-9,9-dimethylxanthene (Xantphos), and 1,10-phenanthroline (Phen), were chosen, given their versatility in organic synthesis, but, in principle, various types of anionic ligands can be encapsulated. The yielded porous supramolecular assemblies (PSAs) were named as PSA-PPh₃, PSA-Xantphos, and PSA-Phen, respectively.

**Physiochemical characterization and local structure analysis**. As a representative material, PSA-PPh₃ was illustrated thoroughly here, whereas the detailed characterization results of other samples were placed in the Supplementary Information (in the section of Supplementary Methods). The chemical composition and local structure of PSA-PPh₃ was determined by solid-state $^{13}C$ and $^{31}P$ NMR spectroscopy. The solid-state $^{13}C$ NMR spectrum of PSA-PPh₃ showed a characteristic signal centered at 40.5 ppm, which is assignable to the opened double bonds formed after polymerization. The chemical shifts centered at 131.3, 147.8, and 154.1 ppm can be ascribed to the non-substituted, substituted, and adjacent aromatic carbons to the S atoms in the polymer backbone, respectively, which are in accord with those of the monomer. The greatly weakened characteristic vinyl $^{13}C$ resonance located in the range of 107.8 to 121.2 ppm suggested the high completion of polymerization (Fig. 2a and see detailed peaks assignments in Supplementary Fig. 8). The solid-state $^{31}P$ NMR spectrum of PSA-PPh₃ gave two distinctive signals at 20.5 and −3.6 ppm, attributable to the phosphonium species from the template and phosphine species from the PPh₃ moieties, respectively (Fig. 2b). These results indicate that the assembly structure is maintained during the polymerization process. The morphology of PSA-PPh₃ was examined by scanning electron microscopy (SEM) and

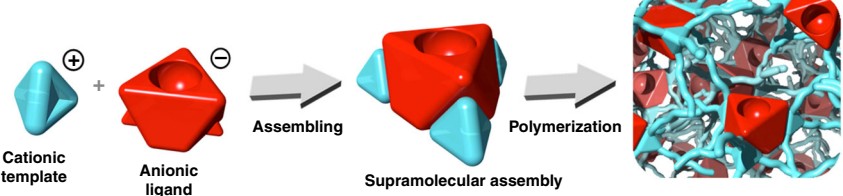

**Fig. 1** Porous supramolecular assemblies. Schematic illustration of the formation of ion pair-directed supramolecular assemblies and their subsequent construction of porous framework

---

**Table 1 Monomer structure and textural parameters of the polymeric assemblies**

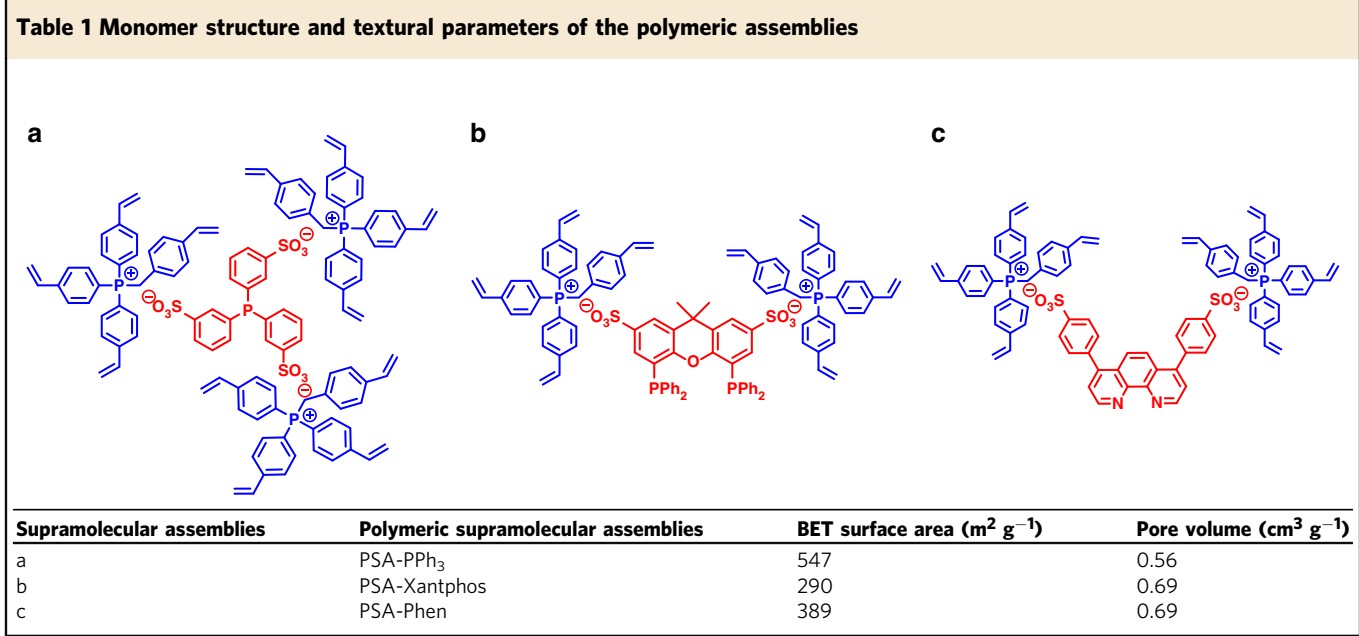

| Supramolecular assemblies | Polymeric supramolecular assemblies | BET surface area (m² g⁻¹) | Pore volume (cm³ g⁻¹) |
|---|---|---|---|
| a | PSA-PPh₃ | 547 | 0.56 |
| b | PSA-Xantphos | 290 | 0.69 |
| c | PSA-Phen | 389 | 0.69 |

---

transmission electron microscopy (TEM), showing that PSA-PPh₃ displayed a rough surface that is agglomerated by irregular particles, giving rise to interconnected mesoporous and macroporous objects (Fig. 2c, d). Analysis of $N_2$ sorption isotherms at 77 K of PSA-PPh₃ revealed a Brunauer–Emmett–Teller (BET) surface area of 547 m² g⁻¹ (Fig. 2e). The pore size distribution, calculated by nonlocal density functional theory modeling, indicated a peak at the lower bound of detection, centered at 1.5 nm, as well as a broad distribution of mesopores (Supplementary Fig. 9). The X-ray powder diffraction pattern showed quite broad peaks, revealing its amorphous nature (Supplementary Fig. 10). Thermogravimetric analysis performed in $N_2$ atmosphere showed the decomposition of the material starting from 400 °C, indicative of its good thermal stability (Supplementary Fig. 11). Remarkably, PSA-PPh₃ displayed excellent amphiphilicity, as demonstrated by the following evidences. When a 1-octene or water droplet was contacted with the surface of PSA-PPh₃ they were momentarily absorbed into the sample, giving rise to swollen surfaces (Supplementary Fig. 12); vapor adsorption experiments revealed that it gave an uptake capacity of 0.43 g g⁻¹ and 0.46 g g⁻¹ for water and toluene vapor, respectively, at a vapor pressure of 3 kPa (Supplementary Fig. 13); liquid adsorption tests showed that it can uptake 8.86 g g⁻¹ water and 5.80 g g⁻¹ 1-octene. Such amphiphilicity is very beneficial to improve the miscibility for the reaction involved with water and oil[43]. To measure the particle size of the polymer in solution, dynamic light scattering (DLS) measurements were performed, showing that the average size of the PSA-PPh₃ particles dispersed in water is in the range of 500 to 1100 nm (Supplementary Fig. 14).

**Evaluation of catalytic performance.** With such a porous supramolecular assembly in hand, we sought to demonstrate its utility toward catalysis by metalation of the embedded ligands. As proof of principle, we chose to react with Rh species, a versatile catalyst applicable for a wide range of organic transformations. The metalation of PSA-PPh₃ using Rh(CO)₂(acac) (acac = acetylacetonato) in toluene afforded an off-white solid (hereafter abbreviated as Rh/PSA-PPh₃, wherein the PPh₃/Rh ratio is 10 if not specifically mentioned). To explore the surface area properties after anchoring of Rh species, $N_2$ sorption isotherms of the Rh/PSA-PPh₃ catalyst was measured, giving similar sorption behavior and a comparable BET surface area as that of PSA-PPh₃ (536 vs. 547 m² g⁻¹, Supplementary Fig. 15). SEM and TEM images showed that there were no appreciable differences between the two materials, confirming the maintenance of the pore structure of PSA-PPh₃ after metalation (Supplementary Fig. 16). Elemental mappings were performed to gain insight into the distribution of Rh species in Rh/PSA-PPh₃. No distinct congregates were observed through high-angle annular dark-field scanning transmission electron microscopy (HAADF-STEM, Supplementary Fig. 17), demonstrating its homogeneous distribution. X-ray photoelectron spectroscopy (XPS) was employed to investigate the coordination environment of Rh species within PSA-PPh₃. The binding energies of Rh3$d_{5/2}$ and Rh3$d_{3/2}$ in Rh/PSA-PPh₃

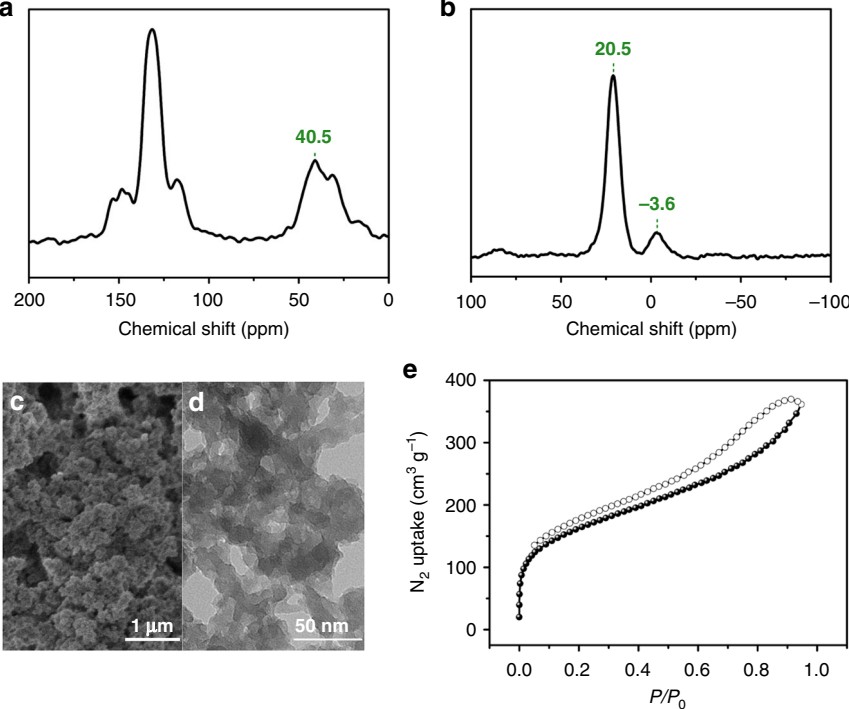

**Fig. 2** Characterization of PSA-PPh₃. (**a**) Solid-state ¹³C NMR, (**b**) solid-state ³¹P NMR, (**c**) SEM image, (**d**) TEM image, (**e**) N₂ sorption isotherms. Source data are provided as a Source Data file

appeared at 308.8 eV and 313.5 eV, respectively, which are lower than those of Rh(CO)₂(acac) (309.3 eV and 314.0 eV). Whereas, the P2p binding energy of the phosphine species in Rh/PSA-PPh₃ (131.8 eV) is higher than that of the parent PSA-PPh₃ (130.9 eV). These results suggest the coordination between the Rh species and PPh₃ moieties in PSA-PPh₃ (Supplementary Fig. 18)[35]. To determine the local coordination environment of the Rh species, IR analyses were conducted, revealing that Rh/PSA-PPh₃ contained the CO (2012 cm⁻¹) and acetylacetone (acac, 1580 cm⁻¹) group. Given that Rh⁺ is a square-planar low-spin d8 metal ion, the corresponding complex is an exception to the 18-electron rule, being a 16-electron complex. Accordingly, we can formulate the structure of Rh species in Rh/PSA-PPh₃ that each Rh ion is coordinated with one CO group, one phosphine ligand, and one acac compound (Supplementary Fig. 19).

To demonstrate the utility of this concept, we targeted hydroformylation given that such transformations greatly facilitate the conversion of a variety of abundant olefins into synthetically versatile aldehyde compounds with 100% atom economy, thereby providing ready access to a large family of important derivatives[44–53]. Among the developed reaction processes, the prominent example is the Ruhrchemie/Rhône-Poulenc process for an aqueous-phase propylene hydroformylation, whereby a rhodium catalyst with water-soluble trisulfonated triphenylphosphane ligand (tppts) was employed[54]. Advantageously, by maintaining the catalyst in the aqueous phase, but not in the emerging product phase, the catalyst and product separation is readily accomplished by aqueous-organic liquid–liquid two-phase extraction. It stands to reason, therefore, that the clear benefits of such processes provide more than enough impetus for performing aqueous-phase transformations. However, the technological challenge of using water as a solvent usually encompasses the reaction of immiscible reagents[55–57]. In the hydroformylation of long-chain alkenes this two-phase extraction technique is not appropriate, given their poor solubility

in the aqueous catalyst phase and thereby limited mass transfer and low efficiency. Several approaches have been taken to increase the reaction rate by enhancing the mobility of the components across the phase boundary. Examples include anchoring the aqueous solution of a water soluble organometallic complex on a high surface area hydrophilic solid; the use of amphiphilic ligands; or performing the reaction in the presence of cyclodextrins, surfactants, and cosolvents such as an alcohol[58–63]. While the reaction rate is significantly promoted, leaching of the catalytic elements into the product phase remains a major drawback. These handicaps associated with the existing catalytic systems prompted us to apply the metallated PSAs in view of their unique structures, featured with a hydrophobic catalytic center and a hydrophilic outer-sphere, and to see the structure-property relationship.

Given the importance of nonanal, the hydroformylation of 1-octene was chosen to demonstrate the viability of our catalytic systems[63–67]. We first contrasted the catalytic activity of Rh/PSA-PPh₃ (the molar ratio of PPh₃ to Rh was optimized at 10, Supplementary Fig. 20) with the well-established Rh/tppts catalyst in aqueous-phase hydroformylation. Using Rh/PSA-PPh₃, a 1-octene conversion of 96.3% with a desired aldehyde products selectivity of 98.4% was achieved within 4 h at 100 °C under a syngas (CO/H₂ = 1/1) pressure of 2.0 MPa. By comparison, Rh/tppts only gave a conversion of 29.7% and aldehyde selectivity of 31.2%, whereby mainly alkene isomers were formed alongside the aldehydes (Table 2, entries 1 and 2). To demonstrate the role of encapsulation and subsequent construction of porous frameworks, control experiments were carried out as delineated in Table 2. The catalytic activities of our benchmark catalytic systems were evaluated with respect to the amount of Rh species (substrate to Rh ratio of 3000), and with a ligand to Rh ratio of 10, if applicable. As expected, the resulting aldehyde proportion is always higher in the presence of the tppts ligand in comparison with that of only Rh(CO)₂(acac). The introduction of the

**Table 2 Catalytic data in the hydroformylation of 1-octene using $H_2O$ as the solvent**

| Entry | Catalyst | Conv.(%) | Aldehydes (%)[a] |
|---|---|---|---|
| 1 | Rh/PSA-PPh$_3$ | 96.3 | 98.4 (n/i = 2.8) |
| 2 | Rh/tppts | 29.7 | 31.2 (n/i = 2.9) |
| 3 | Rh(CO)$_2$(acac) | 26.3 | 24.9 (n/i = 2.8) |
| 4[b] | Rh/tppts + PQP | 32.0 | 52.3 (n/i = 2.8) |
| 5[c] | Rh/tppts + QP | 16.4 | 82.4 (n/i = 2.9) |
| 6[d] | Rh/SA-tppts | 14.4 | 93.8 (n/i = 2.9) |
| 7 | Rh/PSA-Xantphos | 94.7 | 98.0 (n/i = 39) |
| 8 | Rh/2,7-bis(SO$_3$Na)Xantphos | 90.4 | 44.5 (n/i = 3.0) |

*Note*: Reaction conditions: 1-olefin (5.0 mmol), $H_2O$ (10 mL), 100 °C, CO/$H_2$ = 1:1 (2.0 MPa), S/C = 3000, 4 h, and with a ligand to Rh ratio of 10, if applicable. With respect to the entries 7 and 8, the reactions were performed at 120 °C
[a]Selectivity (n/i: molar ratio of linear to branched aldehyde) and other products are iso-olefins
[b]24.8 mg of PQP, synthesized from the polymerization of vinyl-functionalized caionic template, was introduced, wherein the cation moiety is three equiv that of the tppts ligand (NMR results revealed that 46% of tppts remained in the solution)
[c]19.6 mg of QP, synthesized from the reaction between triphenylphosphine and benzyl chloride, was introduced, wherein the cation moiety is three equiv that of the tppts ligand
[d]SA-tppts was synthesized from the ion-exchange between the tppts ligand and QP

polymerized cationic template we used (PQP) and its monomeric analog, benzyltriphenylphosphonium chloride (QP) greatly affected the outcome of the reactions using Rh/tppts, indicating that the phosphonium moieties can alter the reaction. Specifically, the aldehyde selectivity increased from 31.2% to 52.3% and 82.4% for Rh/tppts and that with PQP or QP, respectively (Table 2, entries 3–5). Given the catalytic inactivity of PQP and QP, the observed improved selectivity may be explained by a preferred association of the sodium salt of the sulfonated catalyst complex with the phosphonium moiety containing materials, thereby coupling the catalytically active species with a phase-transfer agent and thus promoting the reaction. Remarkably, the formation of the supramolecular assembly and catalyst encapsulation further engenders reaction selectivity, which reached an even greater extent after construction of the porous framework, as demonstrated by the following experimental evidences. Using the molecular analog of PSA-tppts (SA-tppts) as a modification ligand, gave rise to an aldehyde selectivity of 93.8%, verifying the role of the rational designed outer sphere (Table 2, entry 6). This number was further boosted to 98.4% when the polymeric supramolecular assembly of PSA-PPh$_3$ was employed. In addition, Rh/PSA-PPh$_3$ also far outperformed in terms of activity. We attribute such clear improvements to the effective enrichment of reactants in the pores of framework materials, as evidenced by liquid adsorption tests. It was shown that Rh/PSA-PPh$_3$ can uptake as high as 3.17 g g$^{-1}$ 1-octene, outperforming that afforded by SA-tppts (0.4 g g$^{-1}$) by a factor of 7.9. Moreover, under the catalytic procedure, 35.8 mg of Rh/PSA-PPh$_3$ can absorb 97.8 mg of 1-octene and 72.8 mg of water, confirming the enriched reactant in the catalyst.

The regioselectivity of nonanal, for its part, remains rather constant (in the 2.8–2.9 range) for all catalytic systems evaluated above, suggesting that the encapsulation does not significantly modify the equilibria existing between the rhodium catalytic species. For this particular reaction a combination of high activity and regioselectivity favoring the linear aldehyde is sough-after with regard to the preparation of the subsequent manufacturing (several billion kilograms per year) of plasticizers, soaps, and detergents[63–67]. To increase the linearity, we were interested in encapsulating bidentate phosphine using this established approach. To our delight, an aldehyde yield of 98.0% with a very high n/iso ratio of up to 39 can be achieved using

PSA-Xantphos as a modification ligand (Supplementary Figs. 21–29). Greatly improved performances in terms of both activity and selectivity were achieved compared to all other catalytic systems tested under same reaction conditions (Table 2, entries 7 and 8).

Stability and reusability represent important parameters to evaluate a heterogeneous catalyst for practical applications. Given the superior performance of Rh/PSA-Xantphos in terms of selectivity, it was chosen for detailed study. To assess the heterogeneity of this catalyst system under reaction conditions, we evaluated possible leaching of the catalytic components. Therefore, after the reaction, the catalyst was recovered and the remaining water phase was treated with supplemental monomer. Negligible hydroformylation in the presence of this filtrate solution provides evidence against leaching of a soluble active catalyst. Indeed, it was found that all relevant elements remaining were below the detection limit. Inductively coupled plasma (ICP) analysis of this filtrate solution confirmed low Rh concentration which remained below its detection limit (<10 ppb). $^{31}$P NMR spectrum of the filtrate also revealed that there was no leached Xantphos species (Supplementary Fig. 30). In addition, the catalyst can be readily separated and reused with exactly the same in terms of efficiency and regioselectivity. To further demonstrate the robustness of the catalyst, a successive addition experiment was performed to study the long-term operational properties of Rh/PSA-Xantphos. After completion of the first run, fresh starting material (1-octene, 2.5 mmol) was added directly to the reaction system, and the reaction was monitored as before. The procedure was repeated for six cycles. The conversion of 1-octene for each reaction cycle was constantly near 95% with maintained selectivity, indicating the catalyst is stable during the catalysis and does not undergo major decomposition nor gradually lose activity. This was further validated by the consistent productivity of aldehyde after 2 h over five consecutive runs (Supplementary Table 1). After reaction, the catalyst can be easily recycled through centrifugation or natural settling (Supplementary Fig. 31).

## Discussion
It is worthy of mentioning that the role of water in the Rh/PSAs involved hydroformylation is not limited to easy products isolation, but also promotes the reaction, as evidenced by the

following results. When the reaction was conducted neat, the aldehyde compounds productivity was $0.55 \, \mathrm{mmol \, h^{-1}}$ with an aldehyde selectivity of 70% during the first hour. Further, under otherwise identical conditions, variations of the water amount revealed that the conversion and aldehyde selectivity increased to $0.88 \, \mathrm{mmol \, h^{-1}}$ and 95.3% in the presence of 0.3 mL of water, eventually reaching a plateau of $1.93 \, \mathrm{mmol \, h^{-1}}$ and 96.5% with 10 mL of water, respectively (Supplementary Tables 2–4). This corresponds to a TOF value of $1158 \, \mathrm{h^{-1}}$, placing it among the best materials reported thus far (Supplementary Table 5). Surprisingly, the same reaction conducted using toluene as a solvent, a benchmark system, resulted in a sharp drop in conversion from 96.3% to 55.2%, as well as a decrease in the aldehyde selectivity from 98.4% to 76.3%. When using a mixture of toluene and $H_2O$ (v/v = 1/1) as a reaction medium, the conversion of 1-octene increased to 77.8% (Supplementary Table 3). We reason that water facilitates the formation of rhodium hydride and enhances the hydride transfer step, thereby leading to a higher activity, since most hydroformylation catalysts exhibit a first order dependency on the concentration of this rhodium-hydride species[68,69].

In summary, we have demonstrated an impactful strategy by the utilization of supramolecular chemistry tools to modify the outer sphere environment of each active site in heterogeneous catalysts and consequently, their accompanying reactivity. Catalytic results revealed that this approach provided several layers of tailorability, which can be used to fine-tune the reaction outcomes in terms of both activity and selectivity, as demonstrated in the hydroformylation of olefins. Given the modular design and synthetic simplicity of such directed assembly systems, this strategy can be readily extended to other catalytically active sites. For example, a similar procedure can be used to encapsulate a nitrogen donor, 1,10-phenanthroline (Supplementary Figs. 32–35). This proof-of-concept study is important because it bridges the gap between supramolecular and heterogeneous systems. The strategy should be relevant for parallel and more complex catalytic systems, ultimately accelerating the development of new catalytic materials.

## Methods

**Materials and measurements**. Commercially available reagents were purchased in high purity and used without purification. The purity and structure of the compounds synthesized in this paper were determined by NMR technique. Nitrogen sorption isotherms at 77 K were measured using Micromeritics ASAP 2020 M and Tristar system, after treating the samples under vacuum at 373 °C for 10 h. Porosity distribution were calculated by non-localized density functional theory model: $N_2$ carbon at 77 K based on a slit pores model. Vapor adsorption and desorption isotherms were collected by a BEL sorp-maxII. The intensity-average diameters of the latex particles (Dz) and the polydispersity index (PDI) were measured at 298 K on a ZetaSizer Nano ZS (Malvern Instruments, Worcestershire, U.K.). Thermogravimetric analysis (TG) was performed with a SDT Q600 V8.2 Build 100 instrument in $N_2$ at a heating rate of $10 \, \mathrm{°C \, min^{-1}}$. Powder X-ray diffraction (PXRD) patterns were measured with a Rigaku Ultimate VI X-ray diffractometer (40 kV, 40 mA) using CuKα (λ = 1.5406 Å) radiation. Inductively coupled plasma optical emission spectroscopy (ICP-OES) analysis was measured with a Perkin-Elmer plasma 40 emission spectrometer. The scanning electron microscopy (SEM) images were collected using a Hitachi SU 8010. The transmission electron microscopy (TEM) images were collected using a Hitachi HT-7700. X-ray photoelectron spectra (XPS) of the samples were recorded using a Kratos AXIS Supra with Al Kα X-ray radiations as the X-ray source, and the binding energies were calibrated using C1s peak at 284.9 eV. The solution NMR spectra were recorded with a Bruker Avance-400 spectrometer. Chemical shifts are expressed in ppm downfield from TMS at δ = 0 ppm. $^{13}C$ (100.5 MHz) cross-polarization magic-angle spinning (CP-MAS), and $^{31}P$ (161.8 MHz) MAS solid-state NMR experiments were recorded on a Varian infinity plus 400 spectrometer equipped with a magic-angle spin probe in a 4-mm $ZrO_2$ rotor. The $^{31}P$ NMR chemical shifts were referenced to the $(NH_4)_2HPO_4$. High angle annular dark field-scanning transmission electron microscopy (HADDF-STEM) images were performed using a Titan $G^2$ 80–200 ChemiSTEM FEI. ESI-TOF-MS spectra were recorded on a Shimadzu LC/MS IT-TOF.

## Data availability

The authors declare that all the data supporting the findings of this study are available within the article (and Supplementary Information files), or available from the corresponding author on reasonable request. The source data underlying Fig. 2 is provided as a Source Data file.

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

## Acknowledgements

The authors acknowledge the National Science Foundation of China (21720102001, 91634201, and 21673205), National Key Research and Development Program of China (2017YFB0702803), as well as the National Science Foundation (DMR-1352065) and the University of South Florida for financial support of this work.

## Author contributions

Q.S., S.M. and F.-S.X. conceived and designed the research. K.D., Y.T. and S.W. performed the synthesis and carried out the catalytic tests. C.S. contributed to the analysis of the crystal. B.A. and X.M. provided valuable suggestions for the discussion of the results. All authors participated in drafting the paper, and gave approval to the final version of the paper.

## Additional information

**Competing interests:** The authors declare no competing interests.

