## [Transparent Peer Review File · Nature Communications]

Reviewers' comments:

Reviewer #1 (Remarks to the Author):

The manuscript gives the proof of concept of an innovative catalytic system allowing to combine the benefits of homogeneous catalysis (selectivity in particular) and those of heterogeneous catalysis (easy separation and recycling of the catalyst). It is indeed promising with significant conversion, good selectivity (especially with Xantphos as a ligand) and excellent recyclability. Therefore, in my opinion, the article can be accepted for publication after addressing a few comments.

These supramolecular assemblies also bring together the characteristics of several concepts previously proposed in aqueous biphasic catalysis: namely, the use of amphiphilic ligands, from simple monomers to more evolved structures such as core-shell polymers, and solid-supported aqueous phase catalysis (SAPC), all of which are aimed at confining the hydrophobic substrate in the vicinity of the catalytic complex. The corresponding literature should therefore also be mentioned, in addition to references to cosolvent and other phase-transfer agent agents.

As shown by the authors, the catalysis takes place in the pores of the polymer structure, instead of the bulk aqueous phase; therefore, the role of water might be questioned. In the end, does not it only represent a barrier to the transfer of the reactants to the catalyst surface if the polymer particles are located mainly in the aqueous phase (continuous phase in the reaction tests)? Would these catalysts work in the absence of water or with a very small amount in the manner of SAPC? The catalytic system involving electrostatic interactions, is there any role of the pH of the aqueous phase?

Is the ligand also essentially located in the pores of the solid if the cationic polymer is subsequently added to the aqueous phase containing the TPPTS ligand?

The characterizations of the catalytic system are rather exhaustive; however, the following information would also be useful:

- What is the size of the particles (agglomerates) in solution?
- Does the powder have more affinity for water or organic solvents? (What are the respective contact angles, e.g. obtained by capillary rise (Washburn) or captive bubble method on powder compact?)
- What model was used to determine the pore size distribution by DFT? (cylindrical pore model slit or pore model, etc.)?

Concerning the application in hydroformylation, the following points should be taken into consideration:

- How is the gas phase dispersed in the system? Has it been verified that the resistance to gas-liquid mass transfer is not significant (by increasing the amount of catalyst and / or the stirring speed)?
- How does the 4-phase system behave? Is there a microemulsion between the aqueous phase and the organic phase due to the presence of the particles as in a Pickering emulsion? Do the two liquid phases separate easily?
- Uncertainties on conversion and yield measurement must be given, as well as those from repeatability tests. The TurnOver Frequency (TOF) corresponding to the conversion obtained should be compared with those of other relevant systems from the literature.
- The evolution of catalyst performance during successive recycles should be also detailed as supplementary information. It can be noted that the octene conversion being high (95%) after 4 hours of reaction, the possible loss of catalyst activity during the successive recycles would be better evaluated through the initial reaction rate or the conversion at a shorter time.
- Would it be possible to evaluate the amount of octene adsorbed or embedded within the polymers in the presence of water phase?

Minor corrections: please replace "radial" with "radical" p.5 (2 occurrences).

Reviewer #2 (Remarks to the Author):

Ma, Xiao and co-workers propose an interesting concept for the immobilization of ligands and subsequently organometallic catalysts. The ligands (PPh₃, Xantphos, Phen) are equipped with anionic sulfonate groups and the counteranion is exchanged with a polymerizable tetraphenylphosphonium monomer. Polymerization of the resulting organic salt yields microporous polymer networks, in which the ligand is tightly immobilized by electrostatic interactions. It should be said that parts of this work are not entirely new (see below), however the overall idea is intriguing and I can therefore recommend this work for publication in Nature Commun after the following revisions have been carried out.

1. I'm not happy with the citations. The idea to use electrostatic interactions of a charged framework to immobilize a molecular catalyst with opposite charge has been reported before (Fischer et al. *Angew. Chem. Int. Ed.* 2013, 52, 12174, Chen et al. *Sci. Rep.*, 2015, 5, 11236, Sun et al. *J. Mater. Chem. A*, 2015, 3, 23871). It's especially curious that the last paper was not cited as it is from the authors themselves. Surely, this paper is quite close to the work reported here. However, the catalysts mentioned in these references are prepared by ion exchange within preformed charged networks, so the present work shows indeed an interesting variation of this concept. In addition there are plenty of works which report charged microporous polymer networks for various applications. Within a recent review article on microporous polymers there is a whole chapter on it (Chaoui et al. *Chem. Soc. Rev.*, 2017, 46, 3302). The authors might want to check this again as in their paper there is no distinct reference on charged, ionic networks.
2. It would be interesting to know more about the organic salts used as monomers. Are they crystalline and can a single crystal structure be achieved?
3. A contact angle test like shown in Figure S11 does not make much sense if the liquid is fully absorbed within the porous solid.
4. Why a PPh₃/Rh ratio of 10 was chosen? That seems quite arbitrary. From the experimental part the reader needs to calculate the applied moles of Rh to phosphine on his own (just the weight of the compounds is given), probably the authors could do this for him. So which molar ratio of Rh to PPh₃ was applied? Does all the Rh complex used get immobilized within the network?
5. It's implausible that the network after Rh immobilization still shows the same gravimetric surface area. Even though just any tenth repeating unit is complexed, Rh(xx) adds some weight and volume which can block pores. So is there an explanation, why the surface area here is unchanged?
6. Saying this, how is the Rh actually complexed? There is plenty of analysis for the polymer networks, but not much (XPS, EDX mapping) to the complexed form. So, is the Rh complexed as Ph₃P-Rh(xx) with carbonyl or acac ligand? Both should be easy to see by spectroscopy.
7. The amount of Rh seems not to be enough to make elemental mapping. To conclude that there are no "congregates" seem to me impossible from these low resolution pictures.
8. The catalytic tests look convincing, but I have the feeling that the reaction conditions were actually chosen in such a way, that the immobilized catalyst should show the best performance. Sure, for propylene hydroformylation you might use a two phase system with a sulfonated ligand. However, why adding water to the hydroformylation of 1-octene? How would the conventional Rh catalyst perform in e.g. octene/toluene mixtures?

We greatly appreciate the positive comments and constructive suggestions from the reviewers. We have revised the manuscript accordingly as detailed in the responses below, and the corresponding changes have been highlighted in yellow.

Reviewer #1:

Comment 1: The manuscript gives the proof of concept of an innovative catalytic system allowing to combine the benefits of homogeneous catalysis (selectivity in particular) and those of heterogeneous catalysis (easy separation and recycling of the catalyst). It is indeed promising with significant conversion, good selectivity (especially with Xantphos as a ligand) and excellent recyclability. Therefore, in my opinion, the article can be accepted for publication after addressing a few comments. These supramolecular assemblies also bring together the characteristics of several concepts previously proposed in aqueous biphasic catalysis: namely, the use of amphiphilic ligands, from simple monomers to more evolved structures such as core-shell polymers, and solid-supported aqueous phase catalysis (SAPC), all of which are aimed at confining the hydrophobic substrate in the vicinity of the catalytic complex. The corresponding literature should therefore also be mentioned, in addition to references to cosolvent and other phase-transfer agent agents.

Response: We appreciate the reviewer's high comments and support of our work. The previously developed strategies for promoting the mass transfer in aqueous biphasic catalysis were discussed, and the corresponding references were also adequately cited.

Comment 2: As shown by the authors, the catalysis takes place in the pores of the polymer structure, instead of the bulk aqueous phase; therefore, the role of water might be questioned. In the end, does not it only represent a barrier to the transfer of the reactants to the catalyst surface if the polymer particles are located mainly in the aqueous phase (continuous phase in the reaction tests)? Would these catalysts work in the absence of water or with a very small amount in the manner of SAPC? The catalytic system involving electrostatic interactions, is there any role of the pH of the aqueous phase?

Response: We thank the reviewer for the insightful criticisms. To understand the role of water in the Rh/PSA-PPh₃ catalyzed hydroformylation, a set of control experiments was carried out. When the reaction was conducted neat (20 mmol of 1-octene and 35.8 mg of Rh/PAS-PPh₃ with an S/C = 12000), the aldehyde compounds productivity was 0.55 mmol h⁻¹ with an aldehyde selectivity of 70% during the first hour. Further, under otherwise identical conditions, variations of the water amount revealed that the aldehyde productivity and selectivity increased to 0.88 mmol h⁻¹ and 95.3% in the presence of 0.3 mL of water, eventually reaching a plateau of 1.93 mmol h⁻¹ and 96.5% with 10 mL of water, respectively. These great differences conclude the role of water during the catalytic cycle. We reason that water facilitates the formation of rhodium hydride (catalytically active species) and enhances the hydride transfer

step, thereby leading to a higher activity (Chem. Commun., 2016, 52, 13881). Given the higher aldehyde selectivity when performing the reaction in water, it is suggested that water is beneficial for suppressing the side-reaction of double bond isomerization, which results in the formation of relatively inert internal olefin and thereby leads to a lower aldehyde selectivity.

Per the reviewer's request, the impact of the pH values of the aqueous phase on the Rh/PSA-PPh₃ catalyzed hydroformylation was investigated. Considering the inevitable leaching of catalytic elements from the catalyst at very low or high pH conditions, the catalytic properties of Rh/PSA-PPh₃ at pH 3 and 11 were investigated, whereby leaching of the phosphine ligand was not observed. Variation of the pH of the aqueous solutions led to different results. The conversion of 1-octene decreased from 47.3% at neutral conditions to 38.0% at pH 3, which further dropped to 15.3% at pH 11, suggesting that the pH of the aqueous solution affected the formation of catalytically activity species. The chemo- and regioselectivity of the aldehyde remained rather constant for all the conditions evaluated above (5 mmol of 1-octene, 10 mL of water, 2 MPa of syngas, and 100 °C for 1 h). These data have been included in the revised Supplementary Information (Supplementary Tables 2 and 3).

Comment 3: Is the ligand also essentially located in the pores of the solid if the cationic polymer is subsequently added to the aqueous phase containing the TPPTS ligand?

Response: We thank the reviewer for the comment. To determine the distribution of tppts in the aqueous phase and the cationic polymer, ¹H NMR analysis of the filtration was performed. The content of phosphine species was quantified based on the integration of the internal standard (2,7-bis(SO₃Na)Xantphos). It is shown that with the same catalytic component dosages as we used for catalytic evaluation (11.3 mg of tppts, 24.8 mg of the cationic polymer, and 10 mL of water), 46% of tppts remains in the solution.

Comment 4: The characterizations of the catalytic system are rather exhaustive; however, the following information would also be useful:

- What is the size of the particles (agglomerates) in solution?
- Does the powder have more affinity for water or organic solvents? (What are the respective contact angles, e.g. obtained by capillary rise (Washburn) or captive bubble method on powder compact?)
- What model was used to determine the pore size distribution by DFT? (cylindrical pore model slit or pore model, etc.)?

Response: We thank the reviewer for the valuable comments.

- To measure the particle size of the polymer in solution, dynamic light scattering (DLS) measurements were performed, showing that the average size of the PSA-PPh₃ particles dispersed in water is in the range of 500 to 1100 nm.
- We want to say the polymers are superamphiphilic as evidenced by the following experimental results. It was found that when a 1-octene or water droplet was contacted with the surface of PSA-PPh₃, they were momentarily absorbed into the sample, giving rise to swollen surfaces, thus indicative of its excellent amphiphilicity. This was validated by liquid adsorption tests, showing that PSA-PPh₃ can uptake as high as 8.86 g g⁻¹ water and 5.80 g g⁻¹ 1-octene. To further elucidate the amphiphilicity of PSA-PPh₃, we performed vapor adsorption experiments, showing an uptake capacity of 430 mg g⁻¹ and 673 mg g⁻¹ for water and toluene vapor, respectively, at a vapor pressure of 3 kPa.

To test the contact angle of a liquid on the surface of a sample, we directly drop the corresponding liquid droplet onto the sample in the pressed pellet form. The amount of the liquid is controlled by the instrument, and the photographs of the liquid droplets on the surface are recorded with SL200KB (USA KNO Industry, Co.), according to which the contact angle can usually be measured. In our case, both water and 1-octene droplets are quickly absorbed, giving rise to contact angles of 0°.

- Porosity distribution were calculated by original density functional theory model: N₂ carbon at 77 K based on a slit pores model.

Comment 5: Concerning the application in hydroformylation, the following points should be taken into consideration:

- How is the gas phase dispersed in the system? Has it been verified that the resistance to gas-liquid mass transfer is not significant (by increasing the amount of catalyst and / or the stirring speed)?
- How does the 4-phase system behave? Is there a microemulsion between the aqueous phase and the organic phase due to the presence of the particles as in a Pickering emulsion? Do the two liquid phases separate easily?
- Uncertainties on conversion and yield measurement must be given, as well as those from repeatability tests. The Turn Over Frequency (TOF) corresponding to the conversion obtained should be compared with those of other relevant systems from the literature.
- The evolution of catalyst performance during successive recycles should be also detailed as supplementary information. It can be noted that the octene conversion being high (95%) after 4 hours of reaction, the possible loss of catalyst activity during the successive recycles would be better evaluated through the initial reaction rate or the conversion at a shorter time.
- Would it be possible to evaluate the amount of octene adsorbed or embedded within the polymers in the presence of water phase?

Response: We thank the reviewer for the valuable comments.

- To determine the resistance to the gas-liquid mass transfer, the impact of catalyst amount and the stirring speed on the reaction were investigated. Comparative studies of the variation in stirring speed and catalyst dosage as a function of 1-octene conversion catalyzed by Rh/PSA-PPh₃ clearly showed the disparate rates of hydroformylation. Increasing the catalyst amount from 35.8 mg to 53.7 mg resulted in a marginal increase in conversion from 47.3% to 51.6%, and decreasing the catalyst amount to 17.6 mg gave the 1-octene conversion of 27.6% under otherwise identical conditions. These results are a clear indication of mass transfer limited by the availability of gaseous reagent. This was also evidenced by the fact that decreasing the stirring speed from 600 rpm to 300 rpm led to a great decrease in conversion from 47.3% to 23.4%. Further, decreasing the syngas pressure from 2 MPa to 1 MPa, the 1-octene conversion dropped from 47.3% to 15.8%. We have included these data in the revised Supplementary Information (Supplementary Table 4).
- Due to the amphiphilicity of the catalysts, they stay between the organic and water phase, not in the form of Pickering emulsions. After the reaction, the catalyst can be readily separated through centrifugation or natural settling.
- The catalytic results in the manuscript correspond to the average of at least three different batches, and these results show excellent reproducibility. By taking into account the

precision of GC, the uncertainties of conversion and yield measurement are in the range of $\pm 5\%$. Furthermore, per the reviewer's suggestion, the initial reaction rates and TOF values were estimated to establish a correct comparison with other catalytic systems. To keep the 1-octene conversion below 15% and thereby to ensure the accuracy of TOF values calculation, catalytic reactions were performed under a relatively high substrate to catalyst ratio (S/C = 12000) in the presence of H₂O at 100 °C. Rh/PSA-PPh₃ afforded a TOF value of 1158 h⁻¹ in the first hour. To rank the efficiency, the representative literature results are listed, showing that Rh/PSA-PPh₃ is among the best materials reported thus far (see Supplementary Table 5).

- We thank the reviewer for the valuable criticism. We have evaluated the aldehyde productivity of Rh/PSA-Xantphos after two hours, showing that it can maintain the performance for at least 5 successive recycles (see Supplementary Table 1).
- The absorption capability of the polymers for 1-octene from the water was evaluated as follows: 35.5 mg of Rh/PSA-PPh₃ was introduced into a mixture of water (10 mL) and 1-octene (5 mmol, 561 mg), the same amounts as we used for catalysis. To guarantee the absorptions reached equilibrium, an overnight stirring step was used. After that, the polymer was filtered, and the absorbed 1-octene in Rh/PSA-PPh₃ was quantified by GC using ethylbenzene as an internal standard. Whereby, the amount of the absorbed H₂O was calculated by the following equation: the amount of absorbed H₂O = the increased amount of Rh/PSA-PPh₃ after absorption - the amount of absorbed 1-octene. It is shown that 35.5 mg of Rh/PSA-PPh₃ can absorb 97.8 mg of 1-octene and 72.8 mg of water.

Comment 6: Minor corrections: please replace "radial" with "radical" p.5 (2 occurrences).
We thank the reviewer for pointing this out. We have revised the typo accordingly.

Reviewer #2:

Comment 1: Ma, Xiao and co-workers propose an interesting concept for the immobilization of ligands and subsequently organometallic catalysts. The ligands (PPh₃, Xantphos, Phen) are equipped with anionic sulfonate groups and the counter cation is exchanged with a polymerizable tetraphenylphosphonium monomer. Polymerization of the resulting organic salt yield microporous polymer networks, in which the ligand is tightly immobilized by electrostatic interactions. It should be said that parts of these work are not entirely new (see below), however the overall idea is intriguing and I can therefore recommend this work for publication in Nature Commun after the following revisions have been carried out.

Response: We appreciate the reviewer for taking the time to review our work and providing the valuable comments.

Comment 2: 1. I'm not happy with the citations. The idea to use electrostatic interactions of a charged framework to immobilize a molecular catalyst with opposite charge has been reported before (Fischer et al. Angew. Chem. Int. Ed. 2013, 52, 12174, Chen et al. Sci. Rep., 2015, 5, 11236, Sun et al. J. Mater.Chem. A, 2015, 3, 23871). It's especially curious that the last paper was not cited as it is from the authors themselves. Surely, this paper is quite close to the work reported here.

However, the catalysts mentioned in these references are prepared by ion exchange within preformed charged networks, so the present work shows indeed an interesting variation of this concept. In addition there are plenty of works which report charged microporous polymer networks for various applications. Within a recent review article on microporous polymers there is a whole chapter on it (Chaoui et al. Chem. Soc. Rev., 2017, 46, 3302). The authors might want to check this again as in their paper there is no distinct reference on charged, ionic networks.

Response: We thank the reviewer for pointing these out, and we sincerely apologize for not including the relevant literature citations. Those references have been adequately cited in the revised manuscript.

Comment 3: It would be interesting to know more about the organic salts used as monomers. Are they crystalline and can a single crystal structure be achieved?

Response: We appreciate the reviewer for the valuable comments. In contrast with the cationic template and sulfonic salts, in which single crystals can be readily obtained, the resulting supramolecular assemblies are glass-like amorphous solids (see the pictures below), and our attempts to grow crystals were unsuccessful.

Comment 4: A contact angle test like shown in Figure S11 does not make much sense if the liquid is fully absorbed within the porous solid.

Response: We thank the reviewer for the comment. The purpose of the contact angle tests is to evaluate the wettability of the polymers. Both water and 1-octene droplets can be quickly absorbed into PSA-PPh₃, suggesting its amphiphilicity. This was further validated by liquid adsorption tests and vapor adsorption experiments, revealing that PSA-PPh₃ can uptake 8.86 g g⁻¹ water and 5.80 g g⁻¹ 1-octene, as well as 430 mg g⁻¹ and 460 mg g⁻¹ for water and toluene vapor, respectively, at a vapor pressure of 3 kPa.

Comment 5: Why a PPh₃/Rh ratio of 10 was chosen? That seems quite arbitrary. From the experimental part the reader needs to calculate the applied moles of Rh to phosphine on his own (just the weight of the compounds is given), probably the authors could do this for him. So which molar ratio of Rh to PPh₃ was applied? Does all the Rh complex used was immobilized within the network?

Response: We thank the reviewer for the comments. We have investigated the impact of the PPh₃/Rh ratio on the catalytic performance. It was found that the catalyst with a PPh₃/Rh ratio of 10 gave rise to the optimal results in terms of both activity and selectivity (see Supplementary Figure 20). In addition, per the reviewer's suggestion, we have included the values of applied moles of Rh to phosphine for catalyst synthesis. If not specifically mentioned,

the ratio of PPh_3 to Rh in the catalytic systems investigated was 10. The Rh complex used was fully immobilized within the network as evidenced by the fact that no Rh species was detected in the filtration by ICP-OES. To avoid confusion, we have detailed these pieces of information in the revised manuscript.

Comment 6: It's implausible that the network after Rh immobilization still shows the same gravimetric surface area. Even though just any tenth repeating unit is complexed, $\text{Rh}(\text{xx})$ adds some weight and volume which can block pores. So is there an explanation, why the surface area here is unchanged?

Response: We thank the reviewer for the comment. We ascribe the unchanged surface area to the low Rh species loading amount and the hierarchical pore structure of PSA-PPh_3 . The Rh species weight percent in Rh/PSA-PPh_3 is around 1%, such that the increased weight has relatively little influence on the surface area (or within experimental error). In addition, due to the hierarchical pore structure of the polymer, the concern associated with pore blocking can also be ignored.

Comment 7: Saying this, how is the Rh actually complexed? There is plenty of analysis for the polymer networks, but not much (XPS, EDX mapping) to the complexed form. So, is the Rh complexed as $\text{Ph}_3\text{P-Rh}(\text{xx})$ with carbonyl or acac ligand? Both should be easy to see by spectroscopy.

Response: We thank the reviewer for the valuable comment. To determine the local coordination environment of the Rh species, IR analyses were conducted, revealing that Rh/PSA-PPh_3 contained the CO (2012 cm^{-1}) and acetylacetonate (acac, 1580 cm^{-1}) group. The $\text{Rh}3d$ XPS spectrum of Rh/PSA-PPh_3 indicates that the chemical state of Rh species is +1. Given that Rh^+ is a square-planar low-spin d8 metal ion, the corresponding complex is an exception to the 18-electron rule, being a 16-electron complex. Accordingly, we can formulate the structure of Rh species in Rh/PSA-PPh_3 that each Rh ion is coordinated with one CO group, one phosphine ligand, and one acac compound (see Supplementary Figure 19).

Comment 8: The amount of Rh seems not to be enough to make elemental mapping. To conclude that there are no "congregates" seem to me impossible from this low resolution pictures.

Response: We thank the reviewer for the comment. To answer the reviewer's concern, we have retested these samples. To have more Rh species, a larger mapping area was selected. It is shown that the Rh species were homogeneously distributed in Rh/PSA-PPh_3 and Rh/PSA-Xantphos .

Comment 9: The catalytic tests look convincing, but I have the feeling that the reaction conditions were actually chosen in such a way, that the immobilized catalyst should show the best performance. Sure, for propylene hydroformylation you might use a two phase system with a sulfonated ligand. However, why adding water to the hydroformylation of 1-octene? How would the conventional Rh catalyst perform in e.g. octene/toluene mixtures?

Response: We thank the reviewer for the insightful comments. To understand the role of water in the Rh/PSA-PPh_3 catalyzed hydroformylation, a set of control experiments was carried out. When the reaction was conducted neat (20 mmol of 1-octene and 35.8 mg of Rh/PAS-PPh_3 with an S/C = 12000), the aldehyde compounds productivity was 0.55 mmol h^{-1} with an aldehyde selectivity of 70% during the first hour. Further, under otherwise identical conditions, variations of the water amount revealed that the aldehyde productivity and selectivity increased to 0.88 mmol h^{-1} and 95.3% in the presence of 0.3 mL of water, eventually reaching a

plateau of 1.93 mmol h⁻¹ and 96.5% with 10 mL of water, respectively. Surprisingly, the same reaction conducted using toluene instead of H₂O as a solvent resulted in a sharp drop in conversion from 96.3% to 55.2% as well as a decrease in the aldehyde selectivity from 98.4% to 76.3%. When using a mixture of toluene and H₂O (v/v = 1/1) as a reaction medium, the conversion of 1-octene increased to 77.8%. These great differences conclude the role of water during the catalytic cycle. We reason that water facilitates the formation of rhodium hydride (catalytically active species) and enhances the hydride transfer step, thereby leading to a higher activity (Chem. Commun., 2016, 52, 13881), since most hydroformylation catalysts exhibit a first order dependency in the concentration of this rhodium-hydride species (Rhodium Catalyzed Hydroformylation, ed. P. W. N. M. van Leeuwen and C. Claver, Kluwer Academic Publishers, Dordrecht, 2000; pp.63-105). Given the higher aldehyde selectivity when performing the reaction in water, it is suggested that water is beneficial for suppressing the side-reaction of double bond isomerization, which results in the formation of relatively inert internal olefin and thereby leads to a lower aldehyde selectivity. We have included these data in Supplementary Tables 2 and 3.

Again we thank the reviewers for the constructive comments and suggestions, which have made our manuscript much improved.

REVIEWERS' COMMENTS:

Reviewer #1 (Remarks to the Author):

The authors have conveniently addressed all my comments and queries, performing additional relevant experiments and analyses to this end. Therefore, I consider that the revised version of the manuscript is suitable for publication in Nature Communications. I might only suggest mentioning Supplementary Table 4 (effect of stirring speed and catalyst amount) in the text (note that line 302 actually refers to Supplementary Table 3).

Reviewer #2 (Remarks to the Author):

The authors have answered almost all the concerns of this referee satisfactorily.

- Some missing citations on charged microporous polymer networks are now cited as Ref 22-25, however not at a relevant position, i.e. just after a sentence which generally describes the advantages of immobilized molecular catalysts in porous frameworks. I actually expected that the authors would inform the reader also in the main text that permanently charged microporous networks have been employed before for catalysis and other applications.

- A comment to an answer to Reviewer #1: What is an "Original" DFT model, which is now also stated in the text? NLDFT, QSDFT ?

We greatly appreciate the positive comments and constructive suggestions from the reviewers. We have revised the manuscript accordingly as detailed in the responses below, and the corresponding changes have been highlighted in yellow.

Reviewer #1

Comment: The authors have conveniently addressed all my comments and queries, performing additional relevant experiments and analyses to this end. Therefore, I consider that the revised version of the manuscript is suitable for publication in Nature Communications. I might only suggest mentioning Supplementary Table 4 (effect of stirring speed and catalyst amount) in the text (note that line 302 actually refers to Supplementary Table 3).

We are grateful to the reviewer for taking the time to evaluate our work and support from the reviewer. The concern raised by the reviewer has been addressed.

Reviewer #2

Comment 1: The authors have answered almost all the concerns of this referee satisfactorily.

We thank the reviewer for taking the time to evaluate our work and support from the reviewer.

Comment 2: Some missing citations on charged microporous polymer networks are now cited as Ref 22-25, however not at a relevant position, i.e. just after a sentence which generally describes the advantages of immobilized molecular catalysts in porous frameworks. I actually expected that the authors would inform the reader also in the main text that permanently charged microporous networks have been employed before for catalysis and other applications.

We appreciate the reviewer's constructive suggestion. A brief introduction associated with the permanently charged microporous networks for catalytic applications has been added, and the corresponding references have also been adequately cited.

Comment 3: A comment to an answer to Reviewer #1: What is an "Original" DFT model, which is now also stated in the text? NLDFT, QSDFT?

We thank the reviewer for pointing this out. We use non-localized density functional theory (NLDFT) model.

Again we thank the reviewers for the constructive comments and suggestions, which have made our manuscript much improved.